# Structure-Specific Endonucleases and the Resolution of Chromosome Underreplication

**DOI:** 10.3390/genes10030232

**Published:** 2019-03-19

**Authors:** Benoît Falquet, Ulrich Rass

**Affiliations:** 1Friedrich Miescher Institute for Biomedical Research, Maulbeerstrasse 66, CH-4058 Basel, Switzerland; benoit.falquet@fmi.ch; 2Faculty of Natural Sciences, University of Basel, Petersplatz 10, CH-4003 Basel, Switzerland; 3Genome Damage and Stability Centre, School of Life Sciences, University of Sussex, Falmer, Brighton BN1 9RQ, UK

**Keywords:** DNA replication, chromosome stability, replication stress, Holliday junction resolvase, structure-specific nuclease, ultrafine anaphase bridge, chromosome segregation, mitotic DNA synthesis, genome stability

## Abstract

Complete genome duplication in every cell cycle is fundamental for genome stability and cell survival. However, chromosome replication is frequently challenged by obstacles that impede DNA replication fork (RF) progression, which subsequently causes replication stress (RS). Cells have evolved pathways of RF protection and restart that mitigate the consequences of RS and promote the completion of DNA synthesis prior to mitotic chromosome segregation. If there is entry into mitosis with underreplicated chromosomes, this results in sister-chromatid entanglements, chromosome breakage and rearrangements and aneuploidy in daughter cells. Here, we focus on the resolution of persistent replication intermediates by the structure-specific endonucleases (SSEs) MUS81, SLX1-SLX4 and GEN1. Their actions and a recently discovered pathway of mitotic DNA repair synthesis have emerged as important facilitators of replication completion and sister chromatid detachment in mitosis. As RS is induced by oncogene activation and is a common feature of cancer cells, any advances in our understanding of the molecular mechanisms related to chromosome underreplication have important biomedical implications.

## 1. Introduction

DNA replication requires the unwinding of the parental DNA duplex by the replicative helicase, which leads to the formation of branched DNA structures that are known as replication forks (RFs). Each parental DNA single-strand then acts as a template for DNA synthesis by DNA polymerases, which associate at RFs with large protein assemblies that are known as replisomes. During a human cell cycle, replisomes routinely synthesize DNA with a combined length of approximately two meters, consisting of billions of base pairs. Along the way, the replisomes have to negotiate numerous obstacles, including DNA damage, DNA secondary structures, proteins bound to the DNA template or sites of active transcription. Such obstacles can impede RF progression, causing replication stress (RS). This is mitigated by the replication checkpoint, which activates the pathways for RF recovery and promotes the resumption of DNA synthesis. RF recovery is critically dependent on homologous recombination (HR) and frequently entails the formation of branched DNA intermediates, notably Holliday junctions (HJs) [1,2], which physically link sister chromatids. These HR intermediates are removed by HJ dissolution along a decatenation pathway that is dependent on a complex of Bloom’s syndrome helicase (BLM) and TopoIIIα-RMI1-RMI2 (Sgs1 helicase and Top3-Rmi1 in yeast) [3]. Alternatively, replication-associated HR intermediates can be nucleolytically resolved by a class of structure-specific endonucleases (SSEs) that are known as HJ resolvases [4]. In addition, the cells rely on these same SSEs to cleave chromosomes at sites of potential sister chromatid non-disjunction arising from persistent replication intermediates [5]. This nucleolytic intervention, which leads to chromosome breakage, may appear to be drastic but helps to reinitiate DNA synthesis along HR-dependent repair pathways and serves as a failsafe mechanism for mitotic chromosome segregation. In this review, we provide an overview of the actions of the SSEs Mus81-Mms4/MUS81-EME1 or MUS81-EME2 (budding yeast/human), Slx1-Slx4/SLX1-SLX4 and Yen1/GEN1, highlighting their roles in mitigating genome instability and cell death that results from RS and unfinished DNA replication.

## 2. Intrinsic Safeguards Against Chromosomal Underreplication

Genome replication is a robust process. Eukaryotes have evolved a number of features that help to drive chromosomal replication to completion and minimize the need for SSE interventions [6,7,8]. Chromosomes are subdivided into replication units—or replicons—each initiated at an origin of replication that gives rise to bidirectional RFs. With the exception of the very tips of chromosomes, each inter-origin space is thus replicated by two converging RFs that have adjacent origins. This set-up compensates for local replication shortfalls caused by RF arrest through the actions of oncoming, neighboring RFs. A non-random origin distribution in yeast suggests that the inter-origin distances have been evolutionarily minimized, which reduces the risk of RF double-stalling events (inactivating a pair of converging forks) that may jeopardize replication completion [9]. Secondly, only a fraction of available replication-competent (licensed) origins are normally activated during the S phase of the cell cycle. This overabundance of licensed origins provides cells with a large pool of dormant origins that serve as failsafe replication initiation sites within the areas of insufficient RF progression. The contribution of dormant origins to bulk DNA synthesis is exemplified by the persistence of replication intermediates into the M phase, genome instability and tumor formation in mice upon the experimental depletion of dormant origins and in models with ineffective origin firing [10,11]. Consistently, excess origins have been shown to activate under replication stress conditions, protecting cells from underreplication and DNA damage [12,13,14,15]. Thirdly, the genome of higher eukaryotes is partitioned into multi-replicon replication domains [16]. Origins within a domain activate as a group but with distinct timing from those in other replication domains. This limits the number of active RFs at any one time during S phase, preventing RF destabilization and DNA damage caused by the exhaustion of replication factors or deoxyribonucleoside triphosphate pools (dNTPs) [17,18].

Other safety mechanisms couple DNA replication to cell-cycle progression and ensure that enough time has passed to synthesize a copy of the genome before the cells undergo mitosis. From yeast to humans, mitotic kinase activity is attenuated while DNA replication is ongoing and cells that are unable to initiate DNA replication due to experimental intervention subsequently enter mitosis prematurely [19,20,21,22,23]. This has been linked to a basal activity of the apical checkpoint kinase ATR (Mec1 and Rad3 in *Saccharomyces cerevisiae* and *Schizosaccharomyces pombe*, respectively) in response to single-stranded DNA exposed at active RFs. As a consequence, the expression of the mitotic gene network is suppressed and this avoids premature mitotic entry and carryover of underreplicated DNA into mitosis [24,25,26].

## 3. Preventing Underreplication in the Face of Replication Stress

Replication stress sets off additional cellular pathways that promote full genome replication. While unperturbed replication mildly activates ATR, RS provokes a full-blown ATR response and replication/S-phase checkpoint activation through the exposure of long stretches of RPA-coated single-stranded DNA and single-stranded/double-stranded DNA junctions at stalled RFs. After this, ATR and its orthologues in yeast act with their effector kinases CHK1 and Rad53 in budding yeast and Cds1 in fission yeast to stabilize RFs, upregulate dNTP supplies, modify the DNA replication program and control cell-cycle progression [27,28]. Across organisms, the inhibition of ATR makes cells extremely sensitive to RS and unable to avoid frequent chromosome breakage at intrinsically difficult-to-replicate sites [29,30,31,32,33]. In budding yeast, Mec1-Rad53 signaling prevents RF collapse and promotes stable replication across damaged DNA templates [34,35] while checkpoint disruption results in chromosomal underreplication and accumulation of pathological DNA replication intermediates in the presence of RS [36,37]. Similar observations of RF inactivation and underreplication have been made in vertebrate cells that are acutely deprived of ATR activity [25,26]. RF collapse in yeast and human cells mediated by nucleases and helicases, including Mus81/MUS81, Exo1/EXO1 and SMARCAL1, in the absence of a functional replication checkpoint indicates that the regulation of DNA metabolic enzymes—including SSEs—is one way in which the checkpoint contributes to replication completion under RS conditions [38,39,40,41,42,43]. In addition, origin firing is restrained along the ATR-CHK1 axis across organisms [16,17,44,45,46]. Interestingly, the activation of the replication checkpoint attenuates origin firing globally but the origins at sites of ongoing replication maintain their ability to fire [12,45,46,47]. This limits the number RFs globally when cells experience RS, reducing the risk of excessive RF stalling and DNA-damage formation. At the same time, local origin activation within replicons or replication domains that are already affected by RF blockage promotes replication completion, which further benefits from critical resources (dNTPs, limiting replication factors) not being diverted to sites of newly initiated DNA synthesis in other parts of the genome [48]. Finally, replication checkpoint signaling antagonizes cell-cycle progression by dampening cyclin-dependent kinase (CDK) activity, preventing mitotic entry as long as the unresolved replication problems persist [27,28,49,50,51,52,53,54].

## 4. Structure-Specific Endonucleases and Their Roles in Protecting Cells from Chromosomal Underreplication

Despite the safeguards described above, accidental RF inactivation and collapse are unavoidable and routinely give rise to branched DNA intermediates that require the attention of SSEs. Cells enter the S phase with a large yet finite number of usable replication origins. It follows that double fork-failures affecting pairs of converging RFs without the possibility of compensatory origin firing in the intervening segments of DNA cannot be fully excluded. HR-dependent RF recovery (explained in more detail below) offers possibilities for re-initiating DNA synthesis and SSEs are involved in the timely removal of recombination intermediates that link sister chromatids [5]. On the other hand, theoretical considerations and experimental evidence indicate that the incidence of double-fork failure increases with genome size such that at least one unreplicated genomic site routinely persists until after bulk DNA synthesis in human cells [55,56]. While ongoing replication activity delays mitosis [24,25,26], replication completion appears not to be under stringent checkpoint control and segments of unreplicated DNA can thus be carried forth into mitosis [57]. After this, SSEs can intervene by processing persistent replication intermediates. Their actions resolve DNA entanglements at underreplicated chromosomal segments, which otherwise manifest as ultrafine anaphase bridges (UFBs) between segregating sister chromatids [58,59,60,61]. UFBs are strongly induced by RS and often localize to chromosomal fragile sites (CFSs), which are characterized by a number of features that make them difficult to replicate, such as being transcriptionally active, poor in usable origins and containing repetitive DNA sequences prone to DNA secondary-structure formation [32,33,62]. Fragile site expression—the appearance of metaphase chromosome gaps and breaks—is thought to be the cytogenetic manifestation of extremely late replication and perturbed chromosome condensation at intrinsically difficult-to-replicate, underreplicated chromosomal sites [62,63,64,65,66]. Importantly, fragile sites demarcate the breakpoints of recurrent chromosome rearrangements seen in cancer cells and give rise to deletions and duplications [31,67,68]. Therefore, SSEs play multiple roles in facilitating the completion of genome replication and suppressing genome instability associated with RF failure, incomplete replication and improper chromosome segregation.

## 5. Structure-Specific Endonucleases: Substrate Spectrum and Cell-Cycle Regulation

Mus81-Mms4/MUS81-EME1 or MUS81-EME2, Slx1-Slx4/SLX1-SLX4 and Yen1/GEN1 are three SSEs implicated in removing branched DNA intermediates that arise from stalled and broken RFs in eukaryotes [5] (Figure 1). Their shared ability to cleave DNA four-way junctions places them in the operationally-defined class of HJ resolvases [4]. The resolvases recognize the structure of HJs and catalyze the unique reaction that introduces symmetrically-related incisions across the junction branch point. This reaction completes HR processes by separating the recombining DNA duplexes into nicked duplex products, which can be repaired by simple nick ligation [4]. In contrast to the classic resolvase RuvC from bacteria [69], the eukaryotic HJ resolvases exhibit additional DNA debranching activities on DNA flap structures and DNA three-way junctions that are similar to RFs. Moreover, their actions are tightly regulated by post-translational modifications, protein–protein interactions and nucleocytoplasmic shuttling [70].

### 5.1. MUS81

*MUS81* was identified in a screen for genes that are essential in the absence of *BLM* homologue *SGS1* in budding yeast [73]. The lethality of *sgs1 mus81* double mutant cells was suppressed by inactivating HR [74], which indicates that an accumulation of recombination intermediates arising in the absence of Sgs1-mediated HJ dissolution imposes an essential requirement for Mus81 [73]. Mus81 is a member of the XPF structure-specific endonuclease family and possesses the typical ERCC4 nuclease domain and a pair of terminal helix–hairpin–helix motifs that mediate heterodimer formation with constitutive, non-catalytic subunits [75,76]. These are Mms4 in budding yeast, Eme1 in fission yeast and EME1 or EME2 in vertebrates [77,78,79,80,81,82]. Mus81 complexes from yeast and human were shown to cleave multiple branched DNA substrates, such as DNA 3′-flaps, RFs and HJs. Curiously, recombinant Mus81 consistently showed a clear preference for nicked HJ substrates while cleaving canonical, intact HJs inefficiently [77,83,84]. This is different for MUS81 in complex with EME2, an alternative heterodimeric partner found in vertebrates. The human MUS81-EME2 complex is catalytically more effective than MUS81-EME1 in biochemical assays and can, as a stand-alone nuclease, cleave a wider variety of substrates, including intact HJs and displacements loops (D-loops) generated by HR-mediated strand invasion [85,86] (Figure 1). MUS81-EME2 appears to play a particularly prominent role in the cleavage of RFs [87], which is discussed in more detail below.

Across organisms, Mus81/MUS81 activity is tightly regulated in a cell cycle-dependent manner. In yeast, the catalytic activity of Mus81 is boosted by CDK-dependent hyperphosphorylation of Mms4 (or Eme1 in fission yeast) when the cells approach the G2/M phase of the cell cycle [88,89]. Consistently, 3′-flaps, RFs and nicked HJs were efficiently cleaved by the purified Mus81-Mms4 complex from G2/M cells but not from G1 or S phase-arrested cells [90]. Mms4 hyper-phosphorylation is a multi-step process. First, Cdc5 and Dbf4-dependent kinase (DDK) associate with the scaffold protein Rtt107, which mediates the initial Mms4 phosphorylation together with CDK (Cdc28). This favors the association of Rtt107 and its binding partners with Mms4, providing a positive feedback loop that leads to the hyperphosphorylation of Mms4 when Cdc5 expression peaks towards the end of genome replication [90,91,92]. The cooperation of three kinases acts like a molecular switch, preventing Mus81 activity early in the cell cycle [93] but ensuring robust activation in G2/M. Following mitosis, Mms4 phosphorylation is no longer observed [90] but it is currently unclear whether this is achieved through protein turnover and/or active dephosphorylation. HJ cleavage by Mus81-Mms4 is further modulated by the sumo-like domain protein Esc2 [94] and the proliferating cell nuclear antigen (PCNA) sliding clamp and the clamp loader replication factor C (RFC), which might play a role in recruiting Mus81 to perturbed replication intermediates [95]. Similarly to yeast, the ability of human MUS81 to cleave HJ substrates correlates with the PLK1 and CDK1-dependent phosphorylation of EME1 [88,96]. However, human MUS81 effectively cleaves RF-type substrates at all stages of the cell cycle, indicating that it is not the nuclease activity per se that is cell-cycle regulated [97]. Instead, MUS81-EME1 associates with SLX1-SLX4 and XPF-ERCC1 to form a cell cycle-dependent tri-nuclease complex with HJ resolution activity [96,98,99,100,101] (see Section 5.3 below). The interaction with SLX1-SLX4 seems critical for the recruitment of MUS81 to chromatin during mitosis [97,102].

### 5.2. Slx1-Slx4/SLX1-SLX4

Slx1/SLX1 belongs to the UvrC family of endonucleases with an N-terminal GIY-YIG nuclease domain and a C-terminal zinc-finger domain. Associated with the much larger, multi-domain Slx4/SLX4 protein, Slx1/Slx1 cleaves a variety of DNA substrates, including 5′ flaps, RF analogs and HJs [98,99,100,103,104] (Figure 1). *SLX1* and *SLX4* were uncovered by the same screen for synthetic lethality with *sgs1* that identified *MUS81-MMS4* in budding yeast [73]. In contrast to *mus81 sgs1* cells, the lethality of *slx1 sgs1* or *slx4 sgs1* cells was not suppressed in the absence of HR. It has been proposed that Sgs1 and Slx1-Slx4 cooperate in maintaining the rDNA array in yeast, which might potentially happen by processing stalled RFs to initiate recombinational repair [103,105,106].

In human, SLX1-SLX4 interacts with MUS81-EME1 and XPF-ERCC1 to form the abovementioned tri-nuclease complex that is known as SMX, which functions as a highly effective HJ resolvase [96,98,99,100,101]. Consistently, epistasis analyses place *SLX1*, *SLX4* and *MUS81* in the same pathway of HJ resolution, suppressing sister-chromatid entanglements and mitotic chromosome non-disjunction [107]. However, the expression of SLX4 mutants that are unable to bind MUS81 or SLX1 partially rescues mitotic defects in SLX4-deficient cells [108], pointing to potential additional SLX1 and MUS81-independent roles of SLX4 in the processing of branched DNA intermediates [109]. In human, *SLX4* is one of the genes mutated in Fanconi anemia (and is therefore also known as *FANCP*), a rare genetic disorder characterized by defective repair of replication-blocking inter-strand DNA crosslinks, genome instability, bone marrow failure and a high susceptibility to cancer [110,111].

The crystal structure analyses of *Candida glabrata* Slx1 and the C-terminal region of Slx4 suggest that the formation of inactive Slx1 homodimers provides a means of regulating Slx1-Slx4 complex formation and activity [112]. However, as alluded to above, the control over MUS81 and SLX1-SLX4-dependent HJ resolution has to be considered in the context of cell cycle-dependent SMX complex formation.

### 5.3. The SMX Tri-Nuclease Complex

At its core, the SMX complex has the composite SLX-MUS resolvase that mediates HJ resolution by a SLX1-nick/MUS81-EME1-counternick mechanism [96,100,113]. Co-crystal structures of MUS81-EME1 with DNA have revealed a binding pocket for the 5′-end present at a nick that appears to provide substrate selectivity and enzyme positioning for HJ incision at a point precisely opposite a pre-existing nick [114]. Thus, HJ nicking by SLX1 creates a reference point for HJ resolution by MUS81-EME1, while SLX4 ensures coordinated cleavage by tethering MUS81 and SLX1. These observations provide an explanation for the increased efficiency of four-way DNA junction cleavage upon the association of MUS81-EME1 with SLX1-SLX4 [96,97]. The remaining subunit of SMX, the XPF-ERCC1 heterodimer, stimulates the HJ resolvase activity of SLX-MUS in a manner that is independent of its own nuclease activity [101].

SMX complex formation is governed by the activity of cell-cycle kinases. MUS81 exhibits the highest level of HJ resolution activity when purified from cells arrested in prometaphase by nocodazole at the time when MUS81-EME1 is found to be physically associated with SLX1-SLX4 [88,96]. This protein–protein interaction is dependent on CDK1 and, to a lesser extent, PLK1 activity, and is thus restricted to late cell-cycle phases [96]. CK2-dependent phosphorylation of MUS81 and CDK1-dependent phosphorylation of the SLX4 C-terminal SAP domain have been shown to promote MUS81-SLX4 interactions [72,97,115]. SLX1-SLX4 depletion or ablation of the MUS81 binding domain of SLX4 results in diminished HJ resolution activity of affinity-purified MUS81 or SLX1-SLX4, respectively [96,100]. These findings provide strong evidence that DNA four-way junction cleavage occurs in the context of the SLX-MUS complex in vivo.

In budding yeast, Mus81-Mms4, Slx1-Slx4 and Rad1-Rad10 (the homologue of XPF-ERCC1) have been shown to localize to the same sub-nuclear foci in response to RS and DNA damage [116]. Their localization was not interdependent and did not require the scaffolding function of Slx4, which is consistent with earlier experiments that failed to detect assemblies of a MUS-SLX resolvase in yeast after DNA damage treatment [117]. SSE colocalization occurred in the G1 and S phases and proteins became dispersed upon Mus81-Mms4 activation by hyperphosphorylation in G2/M [116]. These findings suggest that yeast SSEs may be recruited by a common stress-induced signal rather than physical interactions within an SMX complex. However, in a striking parallel to the human system, Mus81-Mms4 has been shown to join the abovementioned complex containing Slx4-Rtt107-Dpb11 as cells enter mitosis [118,119]. Rather than direct binding of Slx4, Mus81-Mms4 recruitment is dependent on a physical interaction between Mms4 and Dpb11, which is mediated by Cdc5 [119]. In contrast to the human system, the Slx4-Dpb11-Mus81-Mms4 complex facilitated the timely resolution of branched DNA intermediates in a Slx1-independent manner and it remains to be determined whether a SLX-MUS-type resolvase is formed in yeast [119,120].

### 5.4. Yen1/GEN1

Yen1 and GEN1 were identified by a two-pronged approach that involved screening the affinity-purified protein complexes from yeast for HJ resolution activity and analyzing HeLa protein fractions with high specific HJ resolution activity by mass spectrometry [121,122]. Yen1/GEN1 are members of the Rad2/XPG nuclease family and possess a bi-partite N-terminal/internal XPG nuclease domain and helix-hairpin-helix domain [123]. While the enzyme is conserved from yeast to humans, it is conspicuously absent in fission yeast, where the heterologous expression of GEN1 can partially substitute for Mus81-Eme1 [71,124,125,126,127,128,129]. Like all other members of the XPG family, Yen1/GEN1 cuts 5′-flap structures but is the only family member that can cleave fully double-stranded three and four-way DNA junctions [121,130] (Figure 1). GEN1 is monomeric in solution and dimerizes on HJs, after which it triggers resolution by dual incision [71,129,131,132,133,134].

In contrast to Mus81-Mms4, Yen1 is inhibited by CDK. Phosphorylated Yen1 resides in the cytoplasm and accumulates in the nucleus after anaphase entry triggers its dephosphorylation by Cdc14. This activates a nuclear import signal and increases the DNA-binding activity of Yen1 [88,135,136,137]. In addition to the regulation by cell cycle-dependent phosphorylation, Yen1 is sumoylated in response to DNA damage. Yen1 sumoylation leads to Slx5-Slx8-dependent ubiquitination and release from DNA. It has been proposed that increased Yen1 turnover mediated by sumoylation limits the mutagenic effects of Yen1 actions on DNA [138].

Nuclear envelope breakdown during mitosis in mammalian cells necessitates a different form of regulation of GEN1 compared to Yen1 in yeast. Strikingly, GEN1 regulation also follows a strategy of cytoplasmic sequestration. A nuclear export signal within GEN1 mediates cytosolic localization throughout interphase (Figure 1), while GEN1 automatically gains access to mitotic chromosomes in prometaphase [139]. GEN1 is phosphorylated in M phase in a similar way to Yen1 although this does not appear to modulate its HJ resolvase activity and the functional consequences of this post-translational modification remain to be determined [139].

## 6. Holliday Junction Resolution by Structure-Specific Endonucleases Facilitates Chromosome Segregation

The canonical function of HJ resolvases is the removal of late HR intermediates. As mentioned above, RF recovery and post-replicative DNA repair pathways rely on HR [140,141] (Figure 2). After this, HJ processing severs any remaining DNA links that may compromise chromosome segregation. The loss of MUS-SLX and Yen1/Gen1-dependent branched-DNA processing sensitizes cells to a variety of agents that impair replication progression by inducing DNA damage and RS. Mus81-defective yeast cells exhibit RS sensitivity, spontaneous chromosome loss, persistence of anaphase-bridge structures and segregation failure and these phenotypes are exacerbated in the absence of Yen1 [119,142,143,144,145,146,147]. Many of the defects can be ameliorated by eliminating HR, which indicates an involvement of unresolved recombination intermediates [136,143,144]. In human cells, the perturbation of the MUS-SLX and GEN1 pathways leads to elevated levels of mitotic chromosome bridges and UFBs, chromosome segregation defects, micronuclei and transmission of DNA damage to daughter cells [96,107,108,148,149]. A recently described UFB sub-type, formed in a manner dependent on HR proteins RAD51 and BRCA2 (termed HR-UFBs), is strongly elevated upon the disruption of MUS81 and GEN1 under RS conditions. This provides evidence that SSE-dependent processing of HR intermediates arising at perturbed RFs is required to ensure that chromosomes are disentangled in time for segregation [150].

Being governed by the regulatory mechanisms described above, which direct the actions of Yen1/GEN1 and Mus81/MUS-SLX towards mitotic chromosomes, HJ resolution occurs late in the cell cycle. Disrupting cell-cycle control over HJ resolution leads to increased crossover formation and loss of heterozygosity from yeast to humans [93,135,139,151]. This can be explained by the fact that HJ cleavage by SSEs produces crossover and non-crossover HR outcomes in equal measure. In contrast, HJ dissolution along the Sgs1/BLM-dependent decatenation pathways always leads to non-crossovers [152]. Thus, delaying the action of SSEs until after bulk DNA synthesis is completed provides a window of opportunity to dissolve—rather than resolve—HR intermediates, preventing sister chromatid exchange, chromosomal translocations (in case of non-allelic recombination) and loss of heterozygosity.

Unscheduled nuclear entry of Yen1 during S phase has been shown to result in replication stress sensitivity [135,151]. Thus, the haphazard processing of DNA replication and repair intermediates is another risk that is associated with SSE activity during S phase. Perhaps the most striking examples of chromosome breakage and genome instability upon SSE dysregulation are observed when CDK1 is prematurely activated by the inhibition of either the G2 checkpoint kinase WEE1 or checkpoint protein CHK1 [153,154,155,156,157,158]. Under these conditions, aberrant SLX-MUS complexes formed in the S phase can trigger a massive cleavage of replicating DNA, which results in a chromosome pulverization phenotype [97]. Restricting HJ resolution to mitosis thus serves a dual purpose of protecting ongoing replication, while ensuring that chromosomal DNA links can be fully removed when segregation is imminent.

## 7. Structure-Specific Endonucleases Cleave DNA Replication Intermediates to Promote Cell Viability

Despite the dangers associated with access of SSEs to replicating chromosomes, evidence has been solidifying in recent years that SSEs target persistent replication intermediates to promote the completion of genome replication. In mouse cells, protracted treatment with DNA replication inhibitors was shown to provoke MUS81-dependent chromosomal breaks that were correlated with replication restart [159,160]. These observations are compatible with the conversion of arrested RFs into transient DNA double-strand breaks, which subsequently serve as substrates for HR-dependent replication restart along the break-induced replication (BIR) pathway [161] (Figure 2). BIR can overcome replication breakdown by rebuilding RFs without the need for fresh origin firing, thus providing an opportunity to complete genome replication at difficult-to-replicate and damaged chromosomal sites. Consistently, MUS81 can promote chromosome breakage, replication restart and viability in human cells suffering various types of endogenous and exogenous replication stress [40,148,149,162,163,164,165,166,167,168]. MUS81-dependent DNA breaks result from alternative complexes containing MUS81-EME1 or MUS81-EME2, with the latter being particularly relevant to RF processing in S-phase cells [87,169]. Interestingly, in contrast to MUS81-EME1, MUS81-EME2 has the ability to process D-loop structures, such as those generated by strand invasion during BIR [86]. This raises the possibility that the actions of MUS81-EME2 may be involved in initiating replication restart by BIR and subsequently in limiting the extent of BIR-associated DNA synthesis. BIR-associated replication is error-prone and at least in yeast, Mus81-Mms4 has been shown to limit the mutagenic effects of BIR [170].

Replication stress is a hallmark of cancer, driving genome instability during tumorigenesis [171]. The involvement of MUS81-EME1 and MUS81-EME2 in RF processing and restart highlights the potential of these and other enzymes involved in RF recovery as possible anti-cancer targets.

## 8. Structure-Specific Endonuclease-Mediated Cleavage of DNA Replication Intermediates Initiates DNA Repair Synthesis in Mitosis

As mentioned above, underreplication gives rise to UFBs and chromosome segregation defects (Figure 3). UFBs, which remain undetected by conventional DNA dyes, are identified by their association with a characteristic set of proteins, including Polo-like kinase 1-interacting checkpoint helicase (PICH; also known as ERCC6-like protein) and BLM [58,59]. In contrast to HR-UFBs [172,173], UFBs believed to result from unreplicated segments of DNA, which are often found associated with CFSs, are flanked by foci of Fanconi anemia protein FANCD2 [60,174]. In early mitosis, a PLK1-dependent SMX complex containing the MUS81-EME1 and XPF-ERCC1 nucleases localizes with FANCD2 on chromosomes, suppresses UFBs and promotes fragile-site expression that is associated with new DNA synthesis [148,149,175]. Based on these observations, a pathway of mitotic DNA synthesis (MiDAS) has been proposed, which resolves persistent replication intermediates in an SSE-dependent manner and initiates repair DNA synthesis when the chromosomes condense in preparation for segregation [175]. MiDAS may be viewed as a last-ditch attempt to complete chromosome replication and a catchall for unreplicated DNA that may escape checkpoint surveillance and pose a serious threat to sister chromatid disjunction and chromosome integrity. In light of this, CFS-associated gaps on mitotic chromosomes are a manifestation of ongoing MiDAS, which locally precludes chromosomal condensation, rather than unrepaired DNA damage [148,149,175] (Figure 3). MiDAS requires HR mediator RAD52 but is inhibited by the strand-exchange recombinase RAD51 [176,177]. RAD52 can catalyze strand annealing and supports BIR at regions bearing small homologies [178,179,180], which suggests that MiDAS represents microhomology-mediated BIR initiated at SSE-generated DNA breaks at arrested RFs [181]. Consistently, MiDAS requires the non-catalytic POLD3 subunit of polymerase δ and involves a conservative mode of DNA replication in a similar way to BIR [175,177]. However, a feature that clearly distinguishes MiDAS from other instances of RF collapse and BIR-dependent replication restart is its apparent dependence on chromosome compaction, making MiDAS a truly mitotic phenomenon. The inhibition of chromosome condensation or stabilization of cohesion on sister-chromatid arms prevented the recruitment of MUS81 and precluded MiDAS [175]. It has been suggested that DNA compaction may expose underreplicated segments of DNA, conceivably facilitating their processing by SSEs [175]. MiDAS is strongly elevated under RS conditions and particularly prevalent in aneuploid cell lines, which makes the pathway an attractive potential target for cancer therapy [175,182].

In yeast, Yen1 has been implicated in the mitotic resolution of underreplicated DNA. Yen1-mutant cells exhibit hypersensitivity to RS upon inactivation of the helicase domain within the Dna2 nuclease-helicase [183,184,185]. This synthetic-sick relationship persisted in the absence of Rad52, indicating that Yen1 targets in Dna2-mutant cells arise independently of HR [185]. In human cells, DNA2 has been shown to promote DNA replication and facilitate the restart of stalled RFs [186,187,188,189]. Consistently, replication intermediates accumulate in Dna2 helicase-defective yeast cells and preclude chromosome segregation unless they are resolved by Yen1 [185]. Due to the fact that Yen1 activity is restricted to mitosis, Dna2-mutant cells are prone to terminal G2/M DNA damage checkpoint arrest when exposed to RS [185]. Conversely, Yen1^ON^, which is a constitutively nuclear and active Yen1 mutant [135], supports the growth of otherwise inviable *dna2*∆ cells [190]. It remains to be determined whether Yen1 cleavage of persistent replication structures results in mitotic DNA synthesis or transmission of DNA breaks to daughter cells. Either way, DNA cleavage of replication intermediates that have escaped Dna2 activity prevents mitotic catastrophe and restores near wild-type levels of viability to Dna2 helicase-mutant cells [185]. In human, the mutations in *DNA2* are associated with Seckel syndrome, one of the microcephalic primordial dwarfism disorders that have been linked to defective RF recovery [191,192]. In cancer, on the other hand, *DNA2* is frequently overexpressed, potentially reflecting an adaptation to endogenous RS and elevated levels of RF stalling [193,194]. If a two-tiered DNA2-GEN1 mechanism for the processing of persistent replication intermediates is conserved in humans, inhibiting these enzymes may provide a means to kill cancer cells by stress overload.

## 9. Structure-Specific Endonuclease Targets Arising at Stalled Replication Forks 

The versatile DNA debranching activities of Mus81-Mms4/MUS81-EME1, MUS81-EME2, Slx1-Slx4/SLX1-SLX4 and Yen1/GEN1 at three-way and four-way DNA junctions suggest that these SSEs may be capable of cleaving a wide variety of failing replication intermediates in vivo. Direct observations of DNA topology by electron microscopy have revealed reversed RF intermediates that are structurally equivalent to HJs [195] (Figure 2). These intermediates are ubiquitous in human cells but accumulate under RS conditions when MUS81 is absent, indicating that SSEs target the remodeled four-way replication structures or arrested RFs that give rise to them [40,196,197].

RF remodeling involving DNA strand separation and strand annealing can be catalyzed by a number of factors, including the DNA helicases and translocases RAD54, HTFL, FBH1, FANCM, SMARCAL1, ZRANB3, BLM and WRN [198]. RF reversal appears to protect from breakage, suggesting that changes in the architecture of forks that were originally perceived as pathologic and detrimental constitute a controlled response to RF stalling [40,165,195,199,200,201]. The regressed arm at reversed RFs has an exposed DNA end and is susceptible to degradation. In human cells, the tumor suppressor BRCA2 promotes the formation of protective RAD51 filaments on reversed RFs, acting in an HR-independent role to block MRE11-mediated nucleolytic degradation and RF demise [196,202,203,204,205,206,207]. Preserving reversed RFs may facilitate passive rescue by fork convergence upon the arrival of a neighboring RF. Alternatively, reversed RFs are converted back to three-way processive forks by the controlled resection of the regressed arm and/or branch migration [43,188,208] (see “direct restart” in Figure 2). Active three-way/four-way structure interconversions mediated by bacterial and viral DNA repair helicases in reconstituted in vitro systems suggest that RF recovery by fork remodeling is a ubiquitous mechanism [209,210]. If direct restart fails, RFs can be restored by cleavage-free HR-dependent mechanisms, involving the invasion of the upstream parental duplex by the regressed arm (see “HR-mediated restart” in Figure 2). However, persistent replication intermediates will eventually become susceptible to SSE cleavage late in the cell cycle, when the HJ resolvases are activated and targeted to chromatin as described above (see “RF cleavage” in Figure 2). This ensures sister chromatid disjunction when chromosome segregation approaches although MiDAS and/or DNA damage repair in daughter cells may subsequently be required for replication completion (Figure 3). It is tempting to speculate that reversed fork structures that are distinct from conventional three-way RFs may attract SSEs in vivo but it is currently unclear whether proper nucleolytic processing is dependent upon prior RF remodeling. It will also be interesting to learn how MUS81-EME2 can be targeted to inactivated RFs in S phase while sparing similar structures that are actively engaged in DNA synthesis. The association of the replisome and other replication/repair factors crowding RFs certainly plays a role in the structural conformation and accessibility of RFs for SSEs.

## 10. Conclusions

In the last few years, there has been considerable progress in our understanding of the function and regulation of SSEs in the resolution of underreplication and sister chromatid non-disjunction. The general picture that has emerged is that SSEs are subject to multiple layers of regulation that largely restrict their activities to mitosis. This serves to protect ongoing replication in S phase and ensures that the persistent chromosomal DNA links are removed in time for chromosome segregation. Incomplete replication may escape checkpoint surveillance but SSEs are mobilized at every mitosis and are ready to avert anaphase-bridge formation and mitotic catastrophe. Perhaps one of the most striking discoveries is the SSE-mediated initiation of DNA synthesis along the MiDAS pathway, identifying a surprisingly late-acting mechanism of replication completion in mitosis. MiDAS provides an appealing solution for maintaining genome stability despite the inevitability of local underreplication during genome replication. The advances that are being made in our detailed understanding of the mechanisms related to RS and chromosomal underreplication will no doubt feed into strategies exploiting cancer-associated RS for new anti-cancer therapeutic approaches.

## Figures and Tables

**Figure 1 genes-10-00232-f001:**
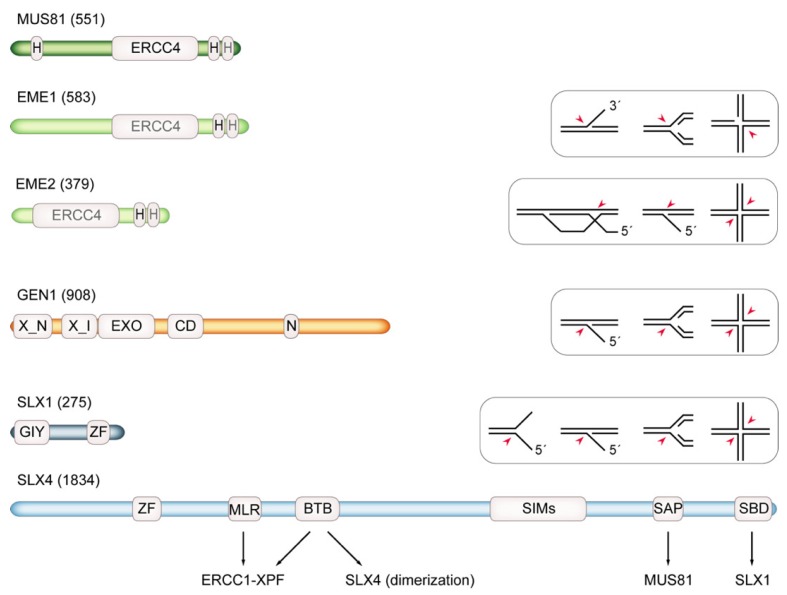
Human structure-specific endonuclease (SSE) domain structures and DNA substrate specificities. MUS81 and its alternative binding partners EME1 and EME2 (length in amino acids is indicated) contain ERCC4 endonuclease domain and helix-hairpin-helix (H) motifs (gray font denotes degenerate motifs). While MUS81-EME1 exhibits activity on 3′-flaps, replication forks (RFs) and nicked Holliday junctions (HJs) (red arrows), MUS81-EME2 additionally cleaves D-loop strand-invasion structures and 5′-flaps as well as being more active on intact HJs. As part of a SLX-MUS complex (see text), MUS81-EME1 effectively resolves HJs by symmetric cleavage after pre-nicking mediated by the SLX1-SLX4 nuclease. GEN1 contains N-terminal and internal XPG nuclease motifs (X_N and X_I), followed by a 5′-3′ exonuclease domain (EXO) and a chromodomain (CD) that promotes substrate recognition [71]; N denotes a nuclear export signal. GEN1 cuts 5′-flaps, RFs and HJs. SLX1 is a GIY-YIG nuclease with a zinc-finger (ZF) at the C-terminus. Associated with SLX4 via a C-terminal SLX1-binding domain (SBD), SLX1 cleaves splayed arm, 5′-flap, RF and HJ substrates. SLX4 contains a ZF domain (two copies of ubiquitin-binding UBZ4), multiple SUMO-interacting motifs (SIMs) [72] and scaffolds a tri-nuclease complex that is known as SMX containing SLX1, MUS81-EME1 (bound at its SAP domain) and ERCC1-XPF (bound via MLR, BTB). Figure is modified from [5].

**Figure 2 genes-10-00232-f002:**
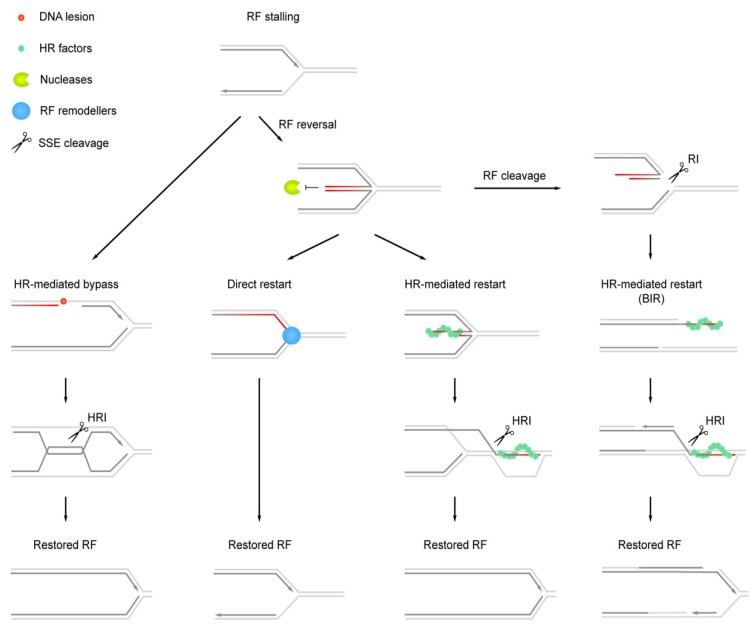
Multiple roles of SSEs in RF recovery. SSEs target replication (RI) and HR (HRI) intermediates to facilitate replication restart and completion. At DNA lesions, RF arrest may be overcome by HR-mediated bypass. Re-initiation of DNA synthesis downstream of lesions leaves daughter-strand gaps that are subsequently filled in by template switching. The ensuing HRIs may be removed by Sgs1/BLM-dependent dissolution (not depicted on figure) or cleavage by SSEs. RF reversal by disengagement of the leading and lagging strands at stalled forks followed by nascent-strand annealing generates HJ-like DNA four-way RIs. These intermediates are shielded from degradation, which facilitates passive rescue by converging RFs. If reversed RFs are not permanently inactivated, such as by replisome loss, remodeling for direct restart that is mediated by DNA helicases/translocases may be possible. Alternatively, functional RFs are restored by HR-mediated restart through invasion of the upstream template and associated HRIs are removed by SSEs. Persistent RIs have emerged as important non-HRI targets of SSEs. The cleavage of RF structures produces single-ended DNA double-strand breaks, triggering break-induced replication (BIR). Invasion of the unbroken sister chromatid generates a D-loop and subsequently a new processive RF. HRIs formed along the BIR pathway are once again resolved by SSEs.

**Figure 3 genes-10-00232-f003:**
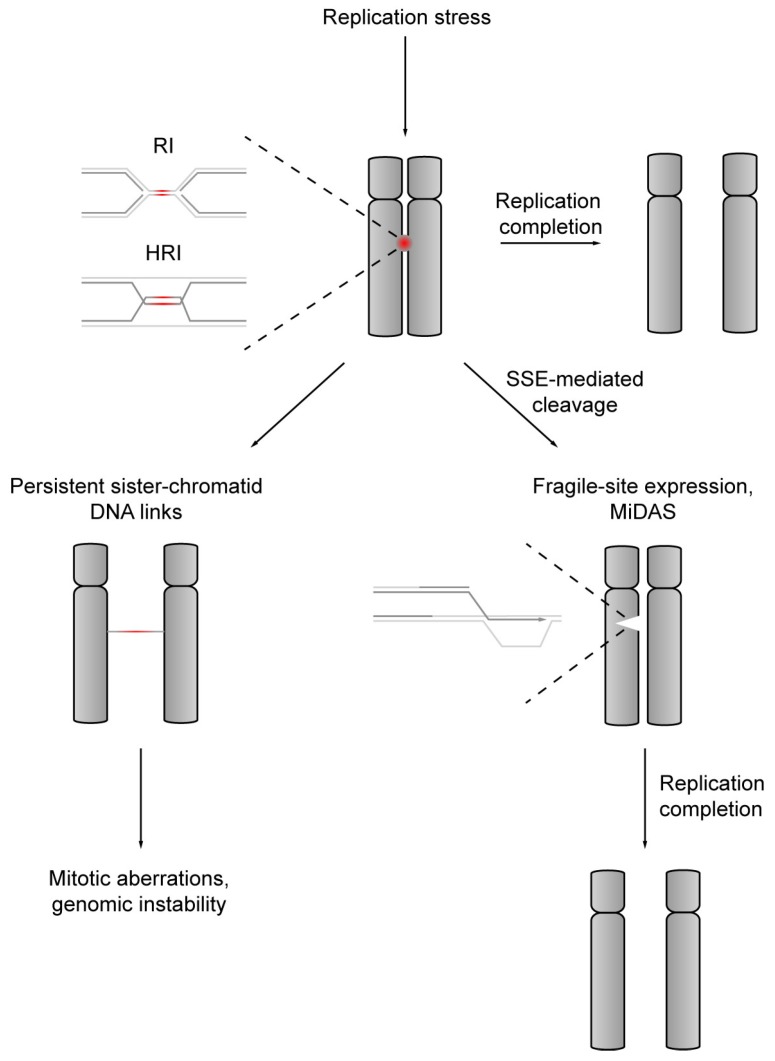
SSEs promote sister chromatid disjunction and replication completion. RS leads to an accumulation of unresolved replication (RI) and HR (HRI) intermediates linking nascent sister chromatids. If replication is not completed in S phase and not all RIs and HRIs are removed, SSEs resolve persistent intermediates in mitosis. RI cleavage initiates late DNA repair synthesis along the mitotic DNA synthesis (MiDAS) pathways, which promotes replication completion in mitotic cells and safeguards sister chromatid disjunction. Failure to resolve RI and HRI sister chromatid DNA links leads to BLM and PICH-bound UFBs, mitotic DNA damage and segregation failure.

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
