# Peer review of "Structure-Specific Endonucleases and the Resolution of Chromosome Underreplication"

_genes, 2019, doi:10.3390/genes10030232_

Round 1

Reviewer 1 Report

This manuscript (MS) entitled “Structure-specific endonucleases and the resolution of chromosome underreplication” reviews the current state of the art of the roles of the main nucleases that deal with replication and recombination intermediates. The MS is nicely written and fully comprehensible, covering an updated vision of the topic and the most relevant and recent literature.

I only have a minor suggestion to improve the MS, could the authors rotate Figure 1 90º clockwise? It would help reading.     

Author Response

This manuscript (MS) entitled “Structure-specific endonucleases and the resolution of chromosome underreplication” reviews the current state of the art of the roles of the main nucleases that deal with replication and recombination intermediates. The MS is nicely written and fully comprehensible, covering an updated vision of the topic and the most relevant and recent literature.

Authors' response: We thank reviewer 1 for the positive feedback on our review MS.

I only have a minor suggestion to improve the MS, could the authors rotate Figure 1 90º clockwise? It would help reading.

Authors' response: Figure 1 (now Figure 2) has been rotated to improve readability.

Reviewer 2 Report

In this review by Falquet & Rass, the role of some structure-specific endonucleases (SSEs) in genome stability maintenance and their regulation during cell cycle is described. The manuscript is clearly written and provides an up-to-date outline of the cellular functions of Mus81-Mms4/MUS81-EME1 or MUS81-EME2, Slx1-Slx4/SLX1-SLX4 and Yen1/GEN1 in budding yeast/human cells. Moreover, the Authors discuss the connection of this SSE set-up with sister chromatid entanglement resolution and mitotic DNA synthesis (MiDAS) pathways, which are critical for proper chromosomal segregation.

However, to help the reader understand and more easily follow the description of the catalytic functions and biological roles of the above SSEs, I would suggest that the Authors add a figure (or a couple of figures) containing a representation of the enzyme polypeptide chains with a map of the relevant protein domains (and, possibly, their post-translational modification sites) together with a schematic drawing of the nucleic acid substrates they are active on.

Minor points:

1. A list of abbreviations used can be added: The acronym SSEs is not specified anywhere in the text. 

2. ZRANB3 is repeated twice in lanes 411-412.

Author Response

In this review by Falquet & Rass, the role of some structure-specific endonucleases (SSEs) in genome stability maintenance and their regulation during cell cycle is described. The manuscript is clearly written and provides an up-to-date outline of the cellular functions of Mus81-Mms4/MUS81-EME1 or MUS81-EME2, Slx1-Slx4/SLX1-SLX4 and Yen1/GEN1 in budding yeast/human cells. Moreover, the Authors discuss the connection of this SSE set-up with sister chromatid entanglement resolution and mitotic DNA synthesis (MiDAS) pathways, which are critical for proper chromosomal segregation.

Authors' response: We thank reviewer 2 for the positive assessment.

However, to help the reader understand and more easily follow the description of the catalytic functions and biological roles of the above SSEs, I would suggest that the Authors add a figure (or a couple of figures) containing a representation of the enzyme polypeptide chains with a map of the relevant protein domains (and, possibly, their post-translational modification sites) together with a schematic drawing of the nucleic acid substrates they are active on.

Authors' response: We thank reviewer 2 for the suggestion. We have added a figure (Figure 1) detailing the domain structure and DNA targets of the HJ-resolving SSEs discussed in the review and agree this is an improvement that makes it much easier to digest the information provided.

Minor points:

1. A list of abbreviations used can be added: The acronym SSEs is not specified anywhere in the text.

Authors' response: The abbreviation SSE is introduced in the Abstract. Because we have kept abbreviations to a minimum, we feel an appendix detailing abbreviations used may not be required.

2. ZRANB3 is repeated twice in lanes 411-412.

Authors' response: We thank reviewer 2 for pointing this out, this has been fixed in the MS.